# GEOCROSSBENCH: CROSS-BAND GENERALIZATION FOR REMOTE SENSING

## ABSTRACT

The number and diversity of remote sensing satellites grows over time, while the vast majority of labeled data comes from older satellites. As the foundation models for Earth observation scale up, the cost of (re-)training to support new satellites grows too, so the generalization capabilities of the models towards new satellites become increasingly important. In this work we introduce GeoCrossBench, an extension of the popular GeoBench benchmark with a new evaluation protocol: it tests the in-distribution performance; generalization to satellites with no band overlap; and generalization to satellites with additional bands with respect to the training set. We also develop a self-supervised extension of ChannelViT, $\chi$ViT, to improve its cross-satellite performance. First, we show that even the best foundation models for remote sensing do not outperform general purpose models like DINOv3 in the in-distribution setting. Second, when generalizing to new satellites with no band overlap, all models suffer 2-4x drop in performance, and $\chi$ViT significantly outperforms the runner-up DINOv3. Third, the performance of all tested models drops on average by 5-25% when given additional bands during test time. Finally, we show that fine-tuning just the last linear layer of these models using oracle labels from all bands can get relatively consistent performance across all satellites, highlighting that the benchmark is far from being saturated. We publicly release the code and the datasets to encourage the development of more future-proof remote sensing models with stronger cross-satellite generalization.

## 1 INTRODUCTION: GENERALIZATION ACROSS REMOTE SENSING DATA

The growth of remote sensing data and satellite imagery in particular (Gorelick et al., 2017; Zhu et al., 2017; Ma et al., 2019) has led to the development of sophisticated deep learning models capable of analyzing complex geospatial patterns and dynamics. Among these, pre-trained foundation models have emerged as a popular paradigm for learning generalizable representations from vast and diverse remote sensing (RS) datasets (Xiong et al., 2024; Fuller et al., 2023; Jakubik et al., 2025; Cong et al., 2022; Han et al., 2024; Tseng et al., 2025; Danish et al., 2025; Jakubik et al., 2023; Wang et al., 2024b). RS data is inherently multimodal, capturing diverse spectral *bands* including multispectral, hyperspectral, and synthetic aperture radar (SAR) (Torres et al., 2012; Drusch et al., 2012; Roy et al., 2014; Guanter et al., 2015). These models promise ease-of-use and transfer across RS data.

While recent foundation models transfer well when train and test bands match, their **cross-band generalization**, to bands and sensors unseen during fine-tuning, remains limited and costly to achieve by retraining. This type of generalization determines how well a model transfers between different spectra and modalities such as from RGB optical to SAR.

This is a critical gap: real-world applications can require models to summarize data from various sensors, to adapt to new spectral bands, or to do a new task that needs bands complementary to the training bands. Robust generalization across spectral domains is crucial for creating more versatile and practical remote sensing models, because large-scale training and fine-tuning is not accessible for all researchers and practitioners.

We introduce **GeoCrossBench** to assess the gap of cross-band generalization in remote sensing with *three* complementary evaluation protocols: (1) *in-distribution* – train and test on the same bands, (2) *no overlap bands generalization* – train on optical bands and test on not overlapping bands, and (3) *superset bands generalization* – test-time inputs provide strictly more bands than used in training.

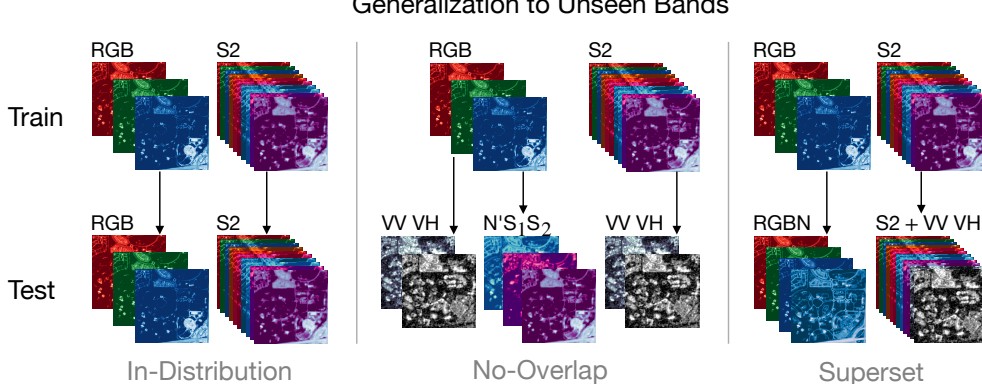

Figure 1: The GeoCrossBench evaluation framework. (1) *In-Distribution*: fine-tune on RGB and evaluate on RGB; fine-tune on full S2 and evaluate on S2. (2) *No-Overlap*: evaluate transfer from RGB→S1 (VV, VH), RGB→N'$S_1S_2$ (B8A, B11, B12) and S2→S1. (3) *Superset*: RGB→RGBN (RGB+NIR) and S2→S2+S1 (optical+SAR fusion).

GeoCrossBench covers three canonical remote sensing tasks: scene classification, semantic segmentation, and change detection, covering both Sentinel-2 (S2) optical/multispectral data and Sentinel-1 (S1) SAR data. Specifically, we build GeoCrossBench from the GeoBench datasets (Lacoste et al., 2023) and enrich them with additional public datasets that widen the range of resolutions and geographic contexts. Moreover, for the datasets missing SAR bands we fuse the Sentinel-2 multispectral bands with co-registered Sentinel-1 SAR bands (VV/VH dual-polarization). This fusion expands the spectral range of the datasets to allow for more rigorous cross-band evaluation. The core idea of GeoCrossBench is to train models on a common band configuration (e.g., RGB, S2) and then evaluate on a variety of unseen bands from both optical and SAR modalities, as illustrated in Figure 1. To provide a comprehensive analysis that also considers practical computational constraints, we evaluate generalization using two primary settings: full fine-tuning and fine-tuning with frozen backbone.

We systematically evaluate a range of existing and recent foundation models using GeoCrossBench. Building on ChannelViT (Bao et al., 2024), an extension of the Vision Transformer (ViT) (Dosovitskiy et al., 2021) for channel-wise modeling, we develop a new baseline for band-wise modeling in RS. We call this model $\chi$**ViT** (ChiViT), short for **Ch**annel-based **i**BOT pre-trained **ViT**, and pretrain it using the iBOT (Zhou et al., 2022) paradigm on our own large-scale, multi-modal dataset.

Experiments with our benchmark reveal insights into current performance and potential directions of improvement. We find that many foundation models struggle with cross-band generalization. Furthermore, we discover that RS-specific foundation models fail to outperform general-purpose vision models like DINOv3 Siméoni et al. (2025) in the in-distribution setting. Finally we show that $\chi$ViT model delivers improved cross-band transfer and achieves best results under these settings. Findings underscore the pressing need for a rigorous and standardized benchmarks like GeoCrossBench.

On publication we will share the GeoCrossBench data, code, and models. This full release can help measure progress, identify weaknesses in current approaches, and ultimately drive the development of more robust, versatile, and reliable foundation models for comprehensive Earth observation.

## 2 GEOCROSSBENCH BENCHMARK: DATASET AND EVALUATION PROTOCOL

GeoCrossBench is designed to thoroughly evaluate the ability of remote sensing foundation models to generalize knowledge learned from one set of spectral bands (specifically RGB or S2) to other spectral band combinations, that either match, do not overlap, or strictly supersets the training bands.

### 2.1 MOTIVATION AND DESIGN PRINCIPLES

The primary motivation behind GeoCrossBench is the observation that many foundation models, despite achieving high performance on tasks when training and testing data come from the same

spectral distribution, but their performance often degrades when processing imagery with different spectral characteristics. This limitation hinders their practical utility. This generalization challenge is not a theoretical concern, it poses a significant barrier to deploying models in a constantly evolving satellite ecosystem where large-scale labeled datasets are often unavailable for newer commercial satellites. A practitioner might need to transfer a model trained on public Sentinel-2 data to newer platforms like Planet SuperDove or Satlantis GARAI, which share some spectral bands with Sentinel-2 but also introduce new ones (e.g., Green I, Yellow). More extreme generalization is required when transferring between entirely different sensor types. For example, adapting an optical model from Sentinel-2 for use with SatVu's HotSat, which captures purely thermal data, requires generalization to a non-overlapping spectral range. The same challenge arises when transferring from optical to SAR imagery. While no benchmark can perfectly replicate every possible transfer task, GeoCrossBench provides a standardized proxy for these real-world challenges, using the best available large-scale labeled data to foster the development of more robust and future-proof foundation models.

GeoCrossBench is built on the following principles:

- **Focus on Generalization:** The main goal is to evaluate how well models adapt from a seen spectral inputs to unseen ones.

- **RS Specific Tasks:** Evaluation is based on tasks central to remote sensing: scene classification, semantic segmentation, and change detection.

- **Diverse Spectral Modalities:** The benchmark incorporates both multi-band optical data and dual-polarization SAR data to test generalization across fundamentally different sensing mechanisms.

## 2.2 DATASETS

GeoCrossBench extends the original GeoBench benchmark by fusing them with corresponding Sentinel-1 SAR data and also incorporates completely new datasets relevant to cross-band generalization, such as x-sen1floods11, x-oscd, x-harvey-flood and x-harvey-building. All datasets in GeoCrossBench utilize Sentinel-2 10-band optical data (B2, B3, B4, B5, B6, B7, B8, B8A, B11, B12 – bands with ≤20m resolution) and Sentinel-1 dual-polarization SAR data (VV, VH – absolute values of the complex numbers), resulting in a 12-band input for each sample. An overview of the datasets is provided in Table 1. The difference between the x-harvey dataset and the original (Rudner et al., 2019) lies in the split we provide, which bypasses the geographical distribution shift. Additionally, we use the original dataset to construct a change detection task by pairing pre- and post-flood images, along with the corresponding flood segmentation masks. x-sen1floods11 is a subset of the original Sen1Floods11 dataset (Bonafilia et al., 2020b), created by removing the weakly labeled portion.

Table 1: Overview of the datasets included in GeoCrossBench. The ones marked with ⋆ are not part of the original GeoBench.

| Dataset Name | Image Size | #Classes | Sensors/Bands | Train | Val | Test |
|---|---|---|---|---|---|---|
| *Classification* | | | | | | |
| x-bigearthnet | $120 \times 120$ | 43 | S2 (10) + S1 (2) | 20000 | 1000 | 1000 |
| x-so2sat | $32 \times 32$ | 17 | S2 (10) + S1 (2) | 19992 | 986 | 986 |
| x-brick-kiln | $64 \times 64$ | 2 | S2 (10) + S1 (2) | 15063 | 999 | 999 |
| x-eurosat | $64 \times 64$ | 10 | S2 (10) + S1 (2) | 2000 | 1000 | 1000 |
| *Semantic Segmentation* | | | | | | |
| x-cashew-plantation | $256 \times 256$ | 7 | S2 (10) + S1 (2) | 1350 | 400 | 50 |
| x-SA-crop-type | $256 \times 256$ | 10 | S2 (10) + S1 (2) | 3000 | 1000 | 1000 |
| x-harvey-building ⋆ | $256 \times 256$ | 2 | S2 (10) + S1 (2) | 375 | 94 | 461 |
| x-sen1floods11 ⋆ | $512 \times 512$ | 2 | S2 (10) + S1 (2) | 252 | 89 | 90 |
| *Change Detection* | | | | | | |
| x-harvey-flood ⋆ | $256 \times 256$ | 2 | S2 (10) + S1 (2) | 375 | 94 | 461 |
| x-oscd ⋆ | $224 \times 224$ | 2 | S2 (10) + S1 (2) | 24 cities | 14 cities | 10 cities |

**Bringing Sentinel-1 data.** For the OSCD dataset (Caye Daudt et al., 2018), we combined it with the corresponding Sentinel-1 data collected by another team (Hafner et al., 2022) to create x-oscd. The m-so2sat dataset (Lacoste et al., 2023; Zhu et al., 2020) from GeoBench already includes paired Sentinel-1 bands. We apply the following transformation to obtain absolute values: $vh = 10 \cdot \log_{10}(vh_i^2 + vh_r^2 + \varepsilon)$, where $\varepsilon = 10^{-10}$, and create x-so2sat. We apply the same transformation to obtain the $vv$ band. For m-eurosat (Lacoste et al., 2023), we retrieve the corresponding Sentinel-1 data from EuroSAT-SAR (Wang et al., 2025) to create x-eurosat. We create x-bigearthnet by pairing m-bigearthnet images (Lacoste et al., 2023) with those from the original set (Sumbul et al., 2021) whose Sentinel-2 parts match those in GeoBench, and then retrieve the corresponding Sentinel-1 images. We create x-cashew-plantation by pairing m-cashew-plantation (Lacoste et al., 2023) with the corresponding Sentinel-1 images retrieved from the Copernicus Open Access Hub (European Space Agency, 2025) using the dates provided in the Sentinel-2 version of the original data. The m-brick-kiln dataset (Lacoste et al., 2023; Lee et al., 2021) does not contain temporal extent information for the imagery. To address this, we collected all available cloud-free Sentinel-2 acquisitions between October 2018 and May 2019. For each sample in m-brick-kiln, we selected a pixel-level similar image from our collected data, recorded its acquisition date, and retrieved the corresponding Sentinel-1 image from the Copernicus Open Access Hub. Using this approach, we constructed x-brick-kiln. We created x-SA-crop-type from the m-SA-crop-type (Lacoste et al., 2023). The original set (Western Cape Department of Agriculture and Radiant Earth Foundation, 2021) contains temporally close Sentinel-1 and Sentinel-2 image pairs, where each Sentinel-1 image was selected as the closest available in time to its corresponding Sentinel-2 image. In m-SA-crop-type, 100 Sentinel-2 images were rotated. To establish accurate matches between Sentinel-1 and Sentinel-2 images, we replaced these rotated images with nearest images from the original dataset.

## 2.3 EVALUATION PROTOCOL

GeoCrossBench evaluates models under three distinct settings designed to probe different aspects of generalization. For all settings, models are fine-tuned on the training data and then evaluated on the test data from various downstream datasets, each representing one of the three remote sensing tasks.

**Setting 1: In-Distribution.** This setting establishes a baseline performance metric. Models are trained and evaluated on the same set of bands. This measures the model's effectiveness in a standard, non-generalization setting. We test two common configurations:

- **Train on RGB → Evaluate on RGB:** Models are fine-tuned using only Sentinel-2's RGB bands (B4, B3, B2).
- **Train on S2 → Evaluate on S2:** Models are fine-tuned using all 10 available Sentinel-2 bands.

**Setting 2: No-Overlap Bands.** This setting tests a model's ability to transfer learned representations to a completely different sensor type, representing a challenging zero-shot generalization task.

- **Train on RGB → Evaluate on S1:** Generalization from RGB to SAR.
- **Train on S2 → Evaluate on S1:** Generalization from multispectral optical to SAR.
- **Train on RGB → Evaluate on N'S₁S₂:** Generalization from RGB to narrow near infrared, shortwave infrared 1 and 2 bands (S2 B8A, B11, B12).

**Setting 3: Superset Bands.** This setting assesses a model's robustness and ability to leverage new information when presented with more spectral bands at test time than it was trained on. This simulates a real-world scenario where a model trained on legacy data must operate on data from a newer, more capable sensor.

- **Train on RGB → Evaluate on RGBN:** Tests generalization from 3-band optical to 4-band optical, adding the Near-Infrared band (S2 B8).
- **Train on S2 → Evaluate on S2+S1:** Tests generalization from 10-band multispectral to a 12-band fused optical-SAR product.

**Scene Classification.** For scene classification tasks, models are trained to assign a class label to an entire image patch. The testing is done by using the evaluation band combination as input and the performance is measured using the corresponding evaluation metric for each task: **F1Score** for x-bigearthnet, **Accuracy** for x-so2sat, x-eurosat and x-brick-kiln.

**Semantic Segmentation.** For semantic segmentation, models are trained to assign a class label to each pixel in an image. At test time, the model segments images using the corresponding band combinations. Segmentation quality is measured by: **mIOU** for x-cashew-plantation, x-SA-crop-type, and x-sen1floods11, and **bIOU** for x-harvey-building.

**Change Detection.** Change detection tasks require the model to identify differences between two remote sensing images of the same area taken at different times. The evaluation involves training on image pairs of the same band combinations and testing on pairs where the 'after' image is replaced with different band combinations, while the 'before' image bands will be kept the same. We report **bIOU** for x-harvey-flood and **F1Score** for x-oscd.

## 3 COMPARISONS: FOUNDATION MODELS, SUPERVISED MODELS, AND A NEW BASELINE

We considered a wide variety of models and fine-tuned them in two primary settings: (i) **full fine-tuning**, where all parameters of the pretrained foundation model and the task-specific head are updated; and (ii) **fine-tuning with frozen backbone**, where only the parameters of a newly added task-specific head (e.g., a linear layer for classification, a decoder for segmentation/change detection) are trained. These settings represent a trade-off between model's training capacity and preservation of the generalization capabilities that might come from pretraining.

### 3.1 PRE-TRAINED FOUNDATION MODELS AND SUPERVISED MODELS

**Specialized Remote Sensing Foundation Models.** We picked most publicly available models pre-trained on remote sensing data having less than 100M parameters (ViT-B and Swin-B), namely TerraMind Jakubik et al. (2025), TerraFM Danish et al. (2025), DOFA Xiong et al. (2024), SatlasNet (Bastani et al., 2023), CROMA Fuller et al. (2023), AnySat Astruc et al. (2024) and Prithvi Jakubik et al. (2023). The details of adapting the bands we need in the benchmarks to the expected inputs of the models are in Appendix E.

**General-purpose Image Foundation Models.** We also added several general-purpose models as baselines. We took self-supervised models of self-distillation type iBOT (Zhou et al., 2022), which is pretrained on ImageNet, DINOv2 (Oquab et al., 2023) and DINOv3 (Siméoni et al., 2025) models pretrained on a huge custom dataset of images. Note, as we are using models having less than 100M parameters, only ViT-B version of DINOv2 and DINOv3 models are used. Moreover, models like the satellite version of DINOv3 (Siméoni et al., 2025) or DINO-MC (Wanyan et al., 2024) lack the ViT-B version and they are not included in our comparison. Following Lacoste et al. (2023), we also fine-tuned ImageNet-pretrained ResNet-50 and ViT-B that have never gone through self-supervised training. Recent work (Xu et al., 2025) has demonstrated that even non-pretrained models can produce competitive results with enough hyperparameter tuning budget. We omitted such baselines as we prefer fine-tuning recipes that are relatively easy and quick to implement for each new downstream task.

**Hyperparameters.** For the original GeoBench tasks, we used a fixed set of hyperparameters selected from prior related works that report their hyperparameters for specific tasks. For segmentation and change detection tasks, we use the UperNet head for all models, which takes the internal representations from 4 layers of the encoder. For change detection, we compute the difference between the encoder representations of the two input images. For the other four tasks, we used a quick hyperparameter search. Refer to Appendix E.1 for the details. Input sizes are chosen to match the size used during pretraining of the underlying model (following (Corley et al., 2024)).

### 3.2 A NEW BASELINE: SELF-SUPERVISED CHANNEL-ViT ON REMOTE SENSING DATA

Robust cross-band generalization requires learning transferable representations from partial spectral inputs. Motivated by recent advances in multi–channel self–supervision we extend ChannelViT (Bao et al., 2024) with a hierarchical pre–training recipe tailored to remote–sensing imagery that we name $\chi$ViT (ChiViT). The core idea is to give each spectral band equal importance during pretraining

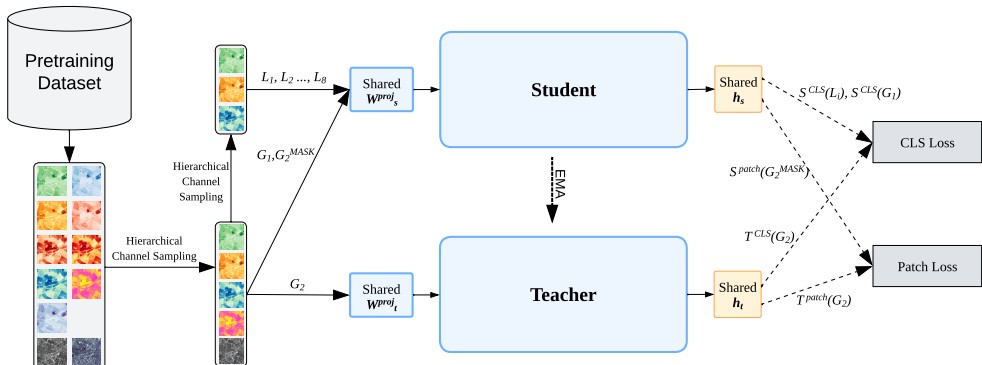

Figure 2: Overview of the iBOT-style self-distillation pretraining used for $\chi$ViT. Hierarchical channel sampling is applied to create distinct views for the student and teacher, where student channels are a subset of the teacher's channels. Shared projection weights and a shared prediction head are utilized, with losses computed for both CLS and patch tokens.

enabling the network to a) fine-tuned on any band subset available without architectural changes and b) exchange information between spectrally distinct modalities.

**Architecture.** ChannelViT preserves channel-specific information by tokenizing each single band independently and adding a learnable *channel embedding* $\mathbf{e}^{\text{chn}}_c$ that is analogous to the positional embedding $\mathbf{e}^{\text{pos}}(h, w)$ of ViT. Given an input $\mathbf{x} \in \mathbb{R}^{C \times H \times W}$, we partition every band into $N_p = (H/P) \cdot (W/P)$ non-overlapping patches of size $P \times P$. Unlike standard ViTs that create a single token from a multi-channel patch, ChannelViT generates one token from each single-channel patch. Thus, for each channel $c \in \{1, ..., C\}$ and each spatial patch $j \in \{1, ..., N_p\}$, we obtain a patch $x_{c,j}$. Each such single-channel patch is flattened into a vector of dimension $P^2$. These flattened patches are then linearly projected into $D$-dimensional embeddings using a learnable linear projection $W \in \mathbb{R}^{P^2 \times D}$. These projection weights $W$ are shared across all channels, promoting the learning of shared low-level features and enhancing robustness (Bao et al., 2024). To retain spatial and channel-specific information, learnable positional embeddings $\mathbf{e}^{\text{pos}}_j \in \mathbb{R}^D$ (shared across channels) and learnable channel embeddings $\mathbf{e}^{\text{chn}}_c \in \mathbb{R}^D$ are added to each projected patch token. A learnable classification token, $\mathbf{e}^{\text{CLS}} \in \mathbb{R}^D$, is prepended to the sequence. The final sequence of $N = C \cdot N_p + 1$ tokens fed to the Transformer encoder is structured as: $[\mathbf{e}^{\text{CLS}}; \ldots; W x_{c,j} + \mathbf{e}^{\text{pos}}_j + \mathbf{e}^{\text{chn}}_c; \ldots]$. This allows the model to reason across both spatial locations and spectral channels simultaneously.

**Pretraining dataset.** To pretrain $\chi$ViT for strong cross-band generalization, we extended Satlas Pretrain dataset (Bastani et al., 2023) up to over 23 million images. This dataset was collected to expose the model to a wide spectrum of Earth's surface characteristics, captured by various spectral bands and resolutions. Notably we added "parallel" data: the BigEarthNet (Sumbul et al., 2021) and Sen12MS datasets (Schmitt et al., 2019), offer Sentinel-1 and Sentinel-2 image pairs that are lined up, crucial for learning joint radar-optical features. Please refer to Appendix C for all details.

Appendix D reports our search over training details.

## 4 GeoCrossBench Experiments and Discussion

Table 2 shows the ranking of all tested models on each evaluation setting. Figure 3 provides a visual summary of key findings.

**RS foundation models do not outperform general-purpose models in-distribution.** Our first key finding from *In-Distribution* setting is that even the best foundation models designed specifically for remote sensing fail to consistently outperform general-purpose vision models like DINOv3. When trained and tested on the same band combinations (e.g., RGB → RGB or S2 → S2), DINOv3 achieves

competitive, and in several cases superior, performance. This suggests that the large-scale, diverse pre-training of general-purpose models provide a powerful and transferable feature foundation that is not yet surpassed by domain-specific pre-training on RS data alone.

**RS foundation models are limited in their cross-band generalization.** The limitations of current foundation models become apparent when generalizing to unseen bands. Under *No-Overlap* setting, which tests generalization to satellites with no band overlap (e.g., S2 → S1), all models suffer a severe 2-4x drop in performance. This highlights a fundamental weakness in transferring learned knowledge across different sensor modalities. To confirm that this performance drop is caused by generalization failure rather than a lack of task-relevant information in the target bands (e.g., SAR), we provide an ablation study on the predictive power of individual bands in Appendix G. However, within this challenging setting, our proposed model, $\chi$ViT, significantly outperforms all other contenders, including the strong runner-up DINOv3.

This weakness is further confirmed by *Superset* setting. Counter-intuitively, providing models with more information at test time by including additional bands also leads to a performance drop, with models degrading by 5-25% on average. This suggests that current architectures may overfit to the specific number and distribution of input channels, failing to robustly integrate novel spectral information without explicit fine-tuning.

**Fine-tuning is often necessary for adequate accuracy.** Full fine-tuning outperforms frozen backbones on average (see Appendix B), with the exception of DINOv2 and TerraFM in the *No-Overlap* scenario (Table 2).

**The value of RS-specific pretraining.** RS-specific pretraining is not delivering top performance on GeoCrossBench against pretraining methods for regular RGB imagery. First, this raises a question: how can these general-purpose models perform transfer at all? One possible explanation is that there are certain correlations between RGB and other bands, especially the features covering the shapes and contours of the objects. The models can learn these patterns from RGB and apply them on other band combinations. Our ablation in Appendix H confirms that a "color-blind" model generalizes better to SAR, highlighting the importance of spatial structure.

Second, one can ask what additional knowledge can RS-specific foundation models learn that will help them beat general-purpose models. The good performance of $\chi$ViT on *No-Overlap* setting hints that additional value can come from careful mixing of images from various satellites during

Table 2: Performance evaluation of all tested models on GeoCrossBench. The * symbol indicates the frozen backbone. The performance metrics of each setting represent the average scores across all GeoCrossBench datasets and the last column indicates the average score across all our settings.

| | In-Distribution | | | | No-Overlap | | | | | Superset | | | | |
|---|---|---|---|---|---|---|---|---|---|---|---|---|---|---|
| **Fine-tuned on Tested on** | RGB RGB | S2 S2 | AVG | # | RGB S1 | S2 S1 | RGB N'S$_1$S$_2$ | AVG | # | RGB RGBN | S2 S2+S1 | AVG | # | Overall AVG |
| $\chi$ViT | 61.81 | 63.53 | 62.67 | 6 | 17.96 | 20.93 | 30.37 | 23.09 | 3 | 59.03 | 57.96 | 58.49 | 1 | 44.51 |
| DINOv3 | 62.46 | 63.0 | 62.73 | 5 | 17.62 | 17.19 | 27.76 | 20.86 | 5 | 52.48 | 59.62 | 56.05 | 2 | 42.88 |
| iBOT | 64.73 | 61.73 | 63.23 | 2 | 18.83 | 14.63 | 26.95 | 20.13 | 8 | 52.36 | 57.04 | 54.7 | 3 | 42.32 |
| TerraMind | 57.78 | 66.32 | 62.05 | 8 | 28.1 | 24.94 | 28.82 | 27.29 | 1 | 49.97 | 34.4 | 42.18 | 15 | 41.48 |
| ViT-B | 62.77 | 62.72 | 62.75 | 4 | 18.87 | 14.75 | 25.18 | 19.6 | 10 | 46.03 | 58.23 | 52.13 | 4 | 41.22 |
| DINOv2 | 65.26 | 62.53 | 63.89 | 1 | 17.36 | 15.01 | 25.85 | 19.41 | 11 | 54.52 | 47.59 | 51.06 | 5 | 41.16 |
| $\chi$ViT* | 56.95 | 58.42 | 57.69 | 13 | 19.02 | 18.92 | 27.12 | 21.69 | 4 | 47.75 | 48.6 | 48.17 | 8 | 39.54 |
| DINOv2* | 61.77 | 56.58 | 59.17 | 10 | 16.28 | 15.84 | 30.13 | 20.75 | 7 | 51.13 | 38.86 | 45.0 | 10 | 38.66 |
| DOFA | 61.71 | 64.39 | 63.05 | 3 | 17.34 | 11.77 | 13.12 | 14.08 | 23 | 50.53 | 48.62 | 49.57 | 6 | 38.21 |
| TerraFM | 61.85 | 62.35 | 62.1 | 7 | 15.9 | 13.53 | 20.82 | 16.75 | 14 | 40.76 | 50.23 | 45.5 | 9 | 37.92 |
| TerraMind* | 52.36 | 61.63 | 57.0 | 14 | 23.26 | 22.38 | 28.74 | 24.79 | 2 | 44.63 | 30.32 | 37.47 | 20 | 37.62 |
| SatlasNet | 49.23 | 69.12 | 59.18 | 9 | 14.62 | 14.61 | 15.4 | 14.88 | 18 | 38.88 | 58.58 | 48.73 | 7 | 37.21 |
| iBOT* | 62.02 | 50.9 | 56.46 | 16 | 15.3 | 13.94 | 30.08 | 19.78 | 9 | 48.63 | 38.13 | 43.38 | 12 | 37.0 |
| ResNet50 | 60.43 | 57.35 | 58.89 | 11 | 13.29 | 13.48 | 19.69 | 15.48 | 16 | 41.53 | 48.33 | 44.93 | 11 | 36.3 |
| DINOv3* | 58.51 | 49.0 | 53.75 | 19 | 14.91 | 17.17 | 30.37 | 20.81 | 6 | 46.69 | 33.54 | 40.11 | 18 | 35.74 |
| DOFA* | 59.1 | 58.03 | 58.57 | 12 | 15.23 | 14.46 | 14.83 | 14.84 | 19 | 38.77 | 45.05 | 41.91 | 16 | 35.07 |
| TerraFM* | 56.7 | 56.72 | 56.71 | 15 | 14.91 | 12.16 | 24.81 | 17.29 | 12 | 40.94 | 35.24 | 38.09 | 19 | 34.5 |
| CROMA | 51.58 | 56.5 | 54.04 | 18 | 16.18 | 12.26 | 16.71 | 15.05 | 17 | 34.04 | 51.61 | 42.82 | 14 | 34.13 |
| ViT-B* | 53.42 | 50.02 | 51.72 | 21 | 16.18 | 14.03 | 21.65 | 17.29 | 13 | 42.87 | 39.84 | 41.35 | 17 | 34.0 |
| Prithvi | 52.61 | 56.88 | 54.74 | 17 | 13.96 | 11.87 | 14.71 | 13.52 | 24 | 32.82 | 53.04 | 42.93 | 13 | 33.7 |
| ResNet50* | 53.13 | 50.22 | 51.68 | 22 | 12.33 | 13.44 | 17.81 | 14.53 | 21 | 41.61 | 31.04 | 36.32 | 21 | 31.37 |
| SatlasNet* | 43.6 | 51.74 | 47.67 | 23 | 12.18 | 13.5 | 13.45 | 13.04 | 25 | 24.9 | 37.17 | 31.04 | 22 | 28.08 |
| CROMA* | 40.99 | 49.7 | 45.34 | 24 | 14.57 | 13.0 | 15.6 | 14.39 | 22 | 20.14 | 37.48 | 28.81 | 23 | 27.35 |
| AnySAT | 47.34 | 58.81 | 53.08 | 20 | 16.19 | 13.79 | 18.3 | 16.09 | 15 | 14.72 | 13.77 | 14.24 | 25 | 26.13 |
| Prithvi* | 43.96 | 28.39 | 36.17 | 26 | 12.68 | 10.04 | 13.29 | 12.0 | 26 | 30.69 | 26.05 | 28.37 | 24 | 23.58 |
| AnySAT* | 40.35 | 47.01 | 43.68 | 25 | 13.35 | 13.57 | 17.52 | 14.81 | 20 | 13.0 | 12.64 | 12.82 | 26 | 22.49 |

pretraining so that the models can learn more complex cross-band relationships between than simple correlations. The poor performance of all tested models in *Superset* setting implies that new ideas are necessary for the models to leverage the additional signal coming from unseen bands at test time. This is a feature that general-purpose models are unlikely to obtain.

**Is the benchmark saturated?**  One way to show that the benchmark is not saturated is to measure the ability of the models to show improved performance when oracle-labeled data is available for other satellites. We examine the frozen representations of the pretrained models. We perform linear probing on the mixture of representation vectors from four band combinations: RGB, Sentinel-2, Sentinel-1 and N'$S_1 S_2$ using the most challenging classification task: x-so2sat. Then we evaluate these linear models on each of the four combinations. We compare them with the linear models trained on only RGB, and on only S2. As seen in Fig. 3b, the performance on S1 can be significantly improved for certain backbones (e.g. $\chi$ViT, DINOv2, DINOv3, TerraFM). This improvement comes with a trade-off: the performance on RGB and S2 slightly decreases with mixture training.

**Implications for future models.**  One of the reasons for the relatively strong performance of $\chi$ViT compared to other multispectral models might be the trick of *sampling of the bands* during pretraining. The models might learn to rely less on band-specific features and instead focus on patterns shared across bands, which then improves cross-band generalization performance. We also tried applying Hierarchical Channel Sampling (HCS) during fine-tuning. Contrary to previous findings (Bao et al., 2024), our experiments on a subset of datasets (see Appendix F) show inconsistent improvements. While HCS aids generalization in specific scenarios, it does not universally improve performance.

The impact of the *scale of the models and datasets* is relatively underexplored in remote sensing. While usually models based on ViT-L outperform similarly trained models based on ViT-B, the usefulness of scaling RS models towards billions of parameters has yet to be demonstrated. While GeoCrossBench limits the number of parameters during the inference, larger models can still be helpful through distillation, e.g. by using techniques demonstrated in DINOv2.

Finally, the *quality and the quantity of pretraining RS data* can have a significant impact on benchmarks like GeoCrossBench. We expect future work prioritize "parallel" imagery datasets. Much like translation data enables knowledge sharing across languages in LLMs, paired images of the same area from different satellites can enhance the cross-band generalization of RS foundation models.

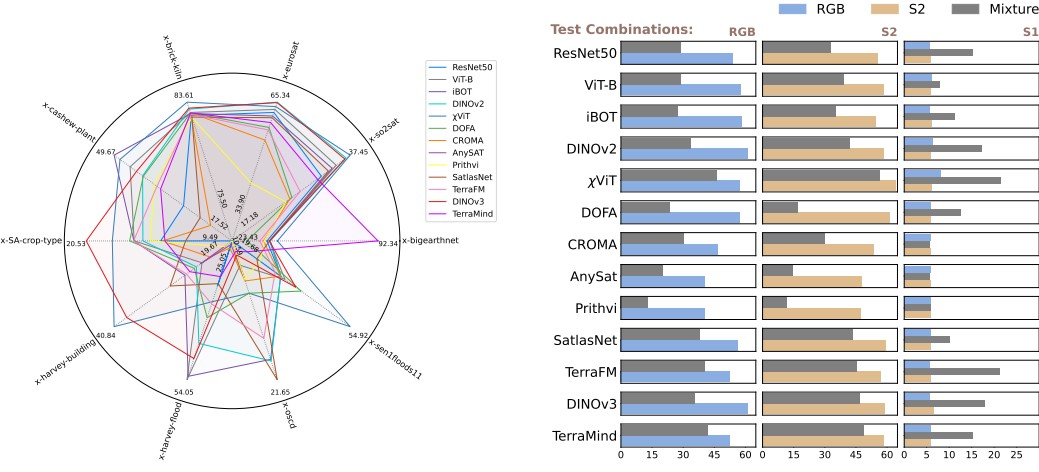

(a) Radar plot for all models and datasets we tried.  (b) Linear probing results on x-so2sat.

Figure 3: **Performance summary on GeoCrossBench.** (a) A radar chart comparing $\chi$ViT against foundation model baselines across the benchmark's tasks. (b) Linear probing on x-so2sat demonstrates that accessing oracle bands (mixture) significantly improves performance on the difficult S1 split.

## 5    RELATED WORK

The development of foundation models for Earth observation has seen rapid progress, aiming to create versatile models applicable to a wide array of downstream tasks.

**Foundation Models for Remote Sensing.**    Remote sensing data is in effect its own data modality, with unique challenges and opportunites, and so there has been a call for specialized machine learning approaches and models (Rolf et al., 2024). This large-scale/small-scale tension motivates foundation modeling to enable more efficiency transfer and application across tasks with limited labels. We highlight pioneering methods that established the topic: SatMAE (Cong et al., 2022) demonstrated self-supervised pre-training for RS at ViT scale and Satlas (Bastani et al., 2023) by contrast demonstrated large-scale and multi-task supervised pre-training for RS. Scale-MAE (Reed et al., 2023) refined self-supervised learning for RS by focusing on different spatial resolutions and generalization across them by controlling for ground sample distance (i.e. the physical size of a pixel). Our $\chi$ViT model and GeoCrossBench dataset is related in spirit to Scale-MAE but spectral, rather than spatial, and explores how to generalize across bands instead of resolutions. GeoCrossBench is a response to the call to do machine learning for remote sensing: it measures the specific need in RS to generalize across bands given the variety of satellites and the varying coverage of data from each.

**Multi-modal/sensor/band Learning in Remote Sensing.**    Multi-modal data with many and different bands is common in remote sensing due to the existence of multiple satellites. As in self-supervised deep learning for other modalities, foundation models in RS learn from multi-modal data in RS: SatMAE (Cong et al., 2022), Scale-MAE (Reed et al., 2023), and MMEarth (Nedungadi et al., 2024) auto-encode multispectral optical data and MMEarth decodes other modalities, while TerraMind (Jakubik et al., 2025) enables any-to-any generation across pixel and token level modalities; SoftCon (Wang et al., 2024b), DeCUR (Wang et al., 2024a), and DOFA (Xiong et al., 2024) separately learn intra-modal representations of multi-spectral and radar data; and CROMA (Fuller et al., 2023), AnySat (Astruc et al., 2024), TerraFM (Danish et al., 2025) and Galileo (Tseng et al., 2025) jointly learn inter-modal representations of multi-spectral and radar data (CROMA) and more modalities like elevation or climate (AnySat, Galileo, TerraMind). In summary these works explore many ways to learn *from* bands but not *across* bands and do not cover how to extend or generalize to new or different bands. GeoCrossBench highlights this direction of improvement, and measures the need for improvement, which is practically motivated by the cost to (re-)train these ever larger foundation models. It is not feasible to train for all combinations of bands, at least not for most groups, so generalization is necessary.

**Datasets and Benchmarking for Remote Sensing.**    Shared datasets and benchmarks are key for comparability in the context of the diversity of RS data and tasks. Our focus is evaluation, like GEO-Bench (Lacoste et al., 2023), and not pre-training, like Terra (Chen et al., 2024). There are many and high-quality task-specific benchmarks (for marine debris (Kikaki et al., 2024), floods (Bonafilia et al., 2020a), agriculture (Garnot and Landrieu, 2021; Rußwurm et al., 2019; Tseng et al., 2021), and more) but they do not focus on general capacities like generalization or efficiency. GeoCrossBench is needed because no existing benchmark measures our key question of cross-band generalization.

### LIMITATIONS AND CONCLUSION

This work focuses on evaluating cross-band generalization capabilities of remote sensing foundation models by extending existing datasets with SAR data. The datasets are limited to static objects and scenes and do not cover moving objects for which it is extremely hard to find parallel optical-SAR imagery. Even if the models achieve perfect scores on our benchmark, they might struggle in detecting moving objects on unseen bands. Our experiments show that RS-specific foundation models have yet to significantly outperform general-purpose models in *No-Overlap* and *Superset* scenarios, and we hope GeoCrossBench will motivate further research in this area.

We have used ChatGPT and Gemini to polish the writing in several sections of this paper.

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

## A  ALL RESULTS

For GeoBench datasets, we report the same metrics as in GeoBench. For other datasets on GeoCross-Bench, we adopt the metrics used in previous works. Specifically, for x-sen1floods11, we report mIoU (Bonafilia et al., 2020b; Marsocci et al., 2024; Tseng et al., 2025). For x-OSCD dataset, we report the F1 score (Caye Daudt et al., 2018; Mendieta et al., 2023). For x-harvey-building and x-harvey-flood, we first calculated mIoU and bIoU (Rudner et al., 2019). However, because the classes are highly imbalanced, our initial experiments showed that models can achieve a high mIoU simply by perfectly segmenting the majority class while completely failing on the minority class. To avoid this misleading result, we therefore report only the bIoU for the minority class as our evaluation metric.

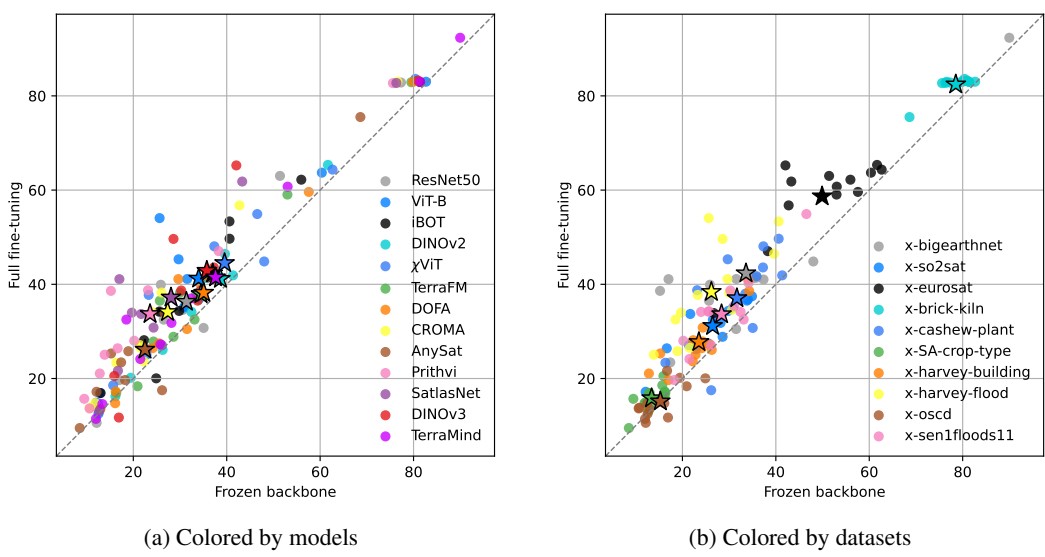

(a) Colored by models                    (b) Colored by datasets

Figure 4: Performance of the models with frozen backbone (x-axis) vs. full fine-tuning (y-axis) for all pairs of models and datasets. (a) figure shows results colored by models, where stars indicate model's average performance. In figure (b) results are colored by datasets and stars are the average performance on each dataset.

## B  FROZEN BACKBONES VS. FULL FINE-TUNING

Figure 4 highlights the difference in performance with frozen and non-frozen backbones. For most of the pairs, full fine-tuning is slightly better. In fact, on average, all models are above the $y = x$ line. Only certain dataset-model pairs are below the diagonal (e.g. AnySat on x-cashew-plant).

## C  PRETRAINING DATASET

The main components are summarized in Table 3. A key part comes from the Satlas Pretrain dataset (Bastani et al., 2023), which includes Sentinel-1 SAR images (using VV and VH polarizations, which are the absolute values of the complex numbers), Sentinel-2 multispectral images (using 10 bands, excluding B01, B09, and B10 which have lower resolution), and NAIP high-resolution RGB aerial photos. The MillionAID dataset (Long et al., 2021) contributes a large volume of RGB aerial images with varied resolutions and image sizes. The BigEarthNet (Sumbul et al., 2021) and Sen12MS datasets (Schmitt et al., 2019), offer Sentinel-1 and Sentinel-2 image pairs that are lined up, crucial for learning joint radar-optical feature. The Intelinair dataset (Chiu et al., 2020) gives very detailed (0.02m GSD) RGB and Near-Infrared (NIR) aerial images of farms. Using all these different datasets together gives $\chi$ViT a solid base for pretraining. This helps the model learn features that work well across different kinds of sensors.

Table 3: Overview of datasets used for $\chi$ViT pretraining.

| Dataset | Bands | # Images | GSD | Image Size |
|---------|-------|----------|-----|------------|
| Satlas (Bastani et al., 2023) (S1) | Sentinel-1 (VV, VH) | 4.5M | 20m | $512 \times 512$ |
| Satlas (Bastani et al., 2023) (S2) | Sentinel-2 (10 bands) | 11M | 10m | $512 \times 512$ |
| Satlas (Bastani et al., 2023) (NAIP) | Aerial RGB | 5.3M | 1m | $512 \times 512$ |
| MillionAID (Long et al., 2021) | Aerial RGB | 2M | 0.5-153m | $(170-550)^2$ |
| BigEarthNet (Sumbul et al., 2021) | S1 SAR & S2 (10 bands) | 0.55M | 10m | $120 \times 120$ |
| Sen12MS (Schmitt et al., 2019) | S1 SAR & S2 (10 bands) | 0.18M | 10m | $256 \times 256$ |
| Intelinair (Chiu et al., 2020) | Aerial RGB, NIR | 34K | 0.02m | $320 \times 320$ |

## D  PRETRAINING DETAILS

The pre-training process is visualized in Figure 2, where we utilized a multi-crop setup with 8 local views (denoted as $L_i$ in the figure) and 2 global views (e.g., $G_1, G_2$). For the final pretraining of $\chi$ViT, we processed 400 million samples in total. The AdamW optimizer (Loshchilov and Hutter, 2019) was used with a batch size of 512. Given that the final pretraining did not have a predefined number of iteration steps, as we aimed to train for as long as we could manage, we utilized the Warmup-Stable-Decay (WSD) learning rate scheduler (Hu et al., 2024). This approach allowed for a flexible decay phase, which was initiated for the last 10% of total iterations. The learning rate was linearly warmed up for the initial 30 million samples to a peak of $2.5 \times 10^{-4}$, maintained during the stable phase, before the final decay. It's common practice to adjust this peak learning rate in proportion to the batch size (e.g., using the formula $peak\_lr \times batch\_size/256$). The overall loss was a sum of the [CLS] token self-distillation loss and the MIM (masked image modeling) loss, without scaling factors between them.

**Design choices.**  To determine the optimal configuration for $\chi$ViT, we conducted several experiments for the key design choices. Each experimental configuration was pre-trained for 40 million samples. Model selection was based on the mean Average Precision (mAP) achieved after fine-tuning on 1% of the BigEarthNet dataset. The results of these ablation studies are summarized in Table 4. Based on these experiments, the winning configuration utilized subset sampling for student channels by sampling from teacher channels (employing hierarchical channel sampling as described in (Bao et al., 2024)), shared projection weights for all bands, a shared prediction head for CLS and patch tokens, and a parallel data coefficient of 4 for the BigEarthNet and Sen12MS datasets during pretraining.

Table 4: Ablation study for $\chi$ViT pretraining design choices. Each configuration was pre-trained for 40M samples. Performance was evaluated by fine-tuning on 1% of BigEarthNet and measuring mAP. PDC refers to the Parallel Data Coefficient.

| Subset Sampling | Shared Proj. | Shared Head | PDC ($\lambda$) | mAP (%) |
|:---:|:---:|:---:|:---:|:---:|
| ✓ | ✓ | ✓ | **4** | **54.72** |
| x | ✓ | ✓ | 4 | 51.10 |
| ✓ | x | ✓ | 4 | 40.44 |
| ✓ | ✓ | x | 4 | 46.55 |
| ✓ | ✓ | ✓ | 8 | 51.51 |

## E  FINE-TUNING

For models whose input channel count is fixed and smaller than the number of bands in our data, we adapt the first convolutional layer. For Sentinel-2 during training, we average the pretrained first-layer weights across the original input channels to obtain a single-channel kernel, replicate this kernel across all input bands, and divide by the new input channel count. For RGBN evaluation with models pretrained for three-channel input, at inference we modify the first layer by setting the weights of the fourth channel to the mean of the weights of the three original channels.

## E.1 Hyperparameters

We apply a grid search to find the best learning rate and decoder depth for the x-sen1floods11, x-harvey-building, x-harvey-flood and x-oscd datasets. Similar to Tseng et al. (2025) we swept learning rates over the sets $\{1, 3, 6\} \times \{10^{-5}, 10^{-4}, 10^{-3}\}$ for full fine-tuning, and $\{1, 3, 4, 5\} \times \{10^{-4}, 10^{-3}, 10^{-2}, 10^{-1}\}$ for fine-tuning with a frozen encoder. Similar to Tseng et al. (2025) we swept learning rates over the sets $\{1, 3, 6\} \times \{10^{-5}, 10^{-4}, 10^{-3}\}$ for full fine-tuning, and $\{1, 3, 4, 5\} \times \{10^{-4}, 10^{-3}, 10^{-2}, 10^{-1}\}$ for fine-tuning with a frozen encoder. For the UPerNet decoder (Xiao et al., 2018), we scale its width with values from the set $\{1, 2, 3\}$ in our grid. Recognizing that the optimal hyperparameters for a given task were often very similar, if not identical, across different models, we conducted the hyperparameter search on selected models, specifically iBOT and DOFA, across all datasets. We then ranked the configurations and selected the top-ranked hyperparameter set that performed well for both iBOT and DOFA, applying them to the remaining models for that particular task. Tables 5 and 6 show the chosen hyperparameters for each dataset.

Table 5: Chosen hyperparameters for full fine-tuning.

| Dataset | Learning rate (LR) | UPerNet width |
|---|---|---|
| x-sen1floods11 | $6 \times 10^{-5}$ | 2 |
| x-harvey-building | $6 \times 10^{-4}$ | 1 |
| x-harvey-flood | $5 \times 10^{-4}$ | 1 |
| x-oscd | $3 \times 10^{-4}$ | 2 |

Table 6: Chosen hyperparameters for fine-tuning with a frozen backbone.

| Dataset | Learning rate (LR) | UPerNet width |
|---|---|---|
| x-sen1floods11 | $4 \times 10^{-4}$ | 2 |
| x-harvey-building | $3 \times 10^{-3}$ | 1 |
| x-harvey-flood | $5 \times 10^{-3}$ | 1 |
| x-oscd | $1 \times 10^{-3}$ | 2 |

For the GeoBench datasets, we adopt fixed hyperparameters without a grid search: we use a learning rate of $1 \times 10^{-4}$ for full fine-tuning and $1 \times 10^{-3}$ when the backbone is frozen. The UPerNet decoder width is set to 3 for all GeoBench tasks.

## E.2 Fine-tuning Details

Across all tasks, we consistently apply the following settings. We use a learning-rate scheduler featuring a 20-epoch linear warmup phase, which is then followed by a cosine decay. This decay period lasts for 30 epochs in classification tasks, and 80 epochs for both segmentation and change detection tasks. We use AdamW optimizer (Loshchilov and Hutter, 2019) for all model trainings. For input normalization, we apply channel-wise mean and standard deviation normalization, clipping the resulting values to the range $[-3, 3]$. Model selection is based on by choosing the checkpoint with the highest validation metric. We set batch sizes to 64 for classification tasks, and to 8 for both segmentation and change detection tasks.

## F Impact of Channel Sampling during Fine-Tuning

To check if the benefits of Hierarchical Channel Sampling (HCS) apply to fine-tuning, we trained $\chi$ViT with HCS on three datasets: x-bigearthnet (classification), x-SA-crop-type (segmentation), and x-harvey-flood (change detection).

The results are presented in Table 7. We observe that the impact of HCS is highly dataset-dependent. On x-bigearthnet, subsampling helps in the *No-Overlap* setting, but on x-SA-crop-type and x-harvey-flood, the improvement is negligible or inconsistent.

Table 7: Comparison of standard fine-tuning versus fine-tuning with Hierarchical Channel Sampling (HCS) using $\chi$ViT. The table reports performance across all three evaluation settings.

| Dataset | Method | In-Distribution | | No-Overlap | | | Superset | |
| | | RGB→RGB | S2→S2 | RGB→S1 | S2→S1 | RGB→N'S$_1$S$_2$ | RGB→RGBN | S2→S2+S1 |
| --- | --- | --- | --- | --- | --- | --- | --- | --- |
| x-bigearthnet | Standard | $71.03 \pm 0.88$ | $73.15 \pm 0.43$ | $5.14 \pm 1.29$ | $6.05 \pm 1.37$ | $19.34 \pm 3.42$ | $68.31 \pm 1.18$ | $70.91 \pm 0.55$ |
| | HCS | $67.39 \pm 0.59$ | $70.85 \pm 0.49$ | $6.83 \pm 2.49$ | $16.95 \pm 4.72$ | $21.88 \pm 13.49$ | $66.12 \pm 0.79$ | $69.49 \pm 1.91$ |
| x-SA-crop-type | Standard | $19.58 \pm 1.28$ | $30.85 \pm 0.54$ | $5.12 \pm 0.09$ | $5.09 \pm 0.02$ | $19.67 \pm 2.16$ | $20.08 \pm 1.17$ | $29.49 \pm 0.42$ |
| | HCS | $24.04 \pm 0.91$ | $30.99 \pm 1.07$ | $5.13 \pm 0.07$ | $5.07 \pm 0.00$ | $23.10 \pm 0.57$ | $24.46 \pm 0.97$ | $29.49 \pm 0.97$ |
| x-harvey-flood | Standard | $64.30 \pm 7.87$ | $56.22 \pm 12.07$ | $4.11 \pm 3.18$ | $10.43 \pm 4.80$ | $18.01 \pm 6.86$ | $59.24 \pm 6.08$ | $51.94 \pm 12.43$ |
| | HCS | $70.66 \pm 0.94$ | $45.72 \pm 10.68$ | $10.59 \pm 6.40$ | $3.64 \pm 1.77$ | $46.90 \pm 2.47$ | $59.70 \pm 2.29$ | $40.98 \pm 8.08$ |

# G  PREDICTIVE POWER OF SPECTRAL BANDS

A critical question in cross-band generalization is whether the target bands contain sufficient physical information to solve the downstream task. This is particularly relevant for vegetation tasks, such as x-cashew-plantation and x-SA-crop-type, where biomass estimation heavily relies on the interplay between Red and Near-Infrared (NIR) channels.

To distinguish between the lack of information in the signal and the model's failure to generalize, we evaluated the "upper bound" performance of two representative models (DOFA and DINOv3) in an in-distribution setting. We trained and evaluated the models specifically on the target band subsets: RGB, N'S$_1$S$_2$ (NIR/SWIR), and S1 (SAR).

The results, presented in Table 8, demonstrate two key findings:

1. **Information Content:** While Sentinel-1 (S1) data contains less predictive information than the full Sentinel-2 spectrum (achieving $\sim$20% lower mIoU on x-cashew-plantation), it still retains significant discriminative power. For example, DINOv3 trained and tested on S1 achieves 67.91 mIoU on the cashew dataset.

2. **Generalization Gap:** There is a massive disparity between the information available in the target bands and the ability of models to access it via transfer. On x-cashew-plantation, the transfer performance from RGB → S1 ($\sim$6.78 mIoU) is an order of magnitude lower than the S1 → S1 baseline (67.91 mIoU).

This confirms that the poor performance reported in the *No-Overlap* setting of GeoCrossBench is not due to the physical irrelevance of the target bands, but rather the foundation models' inability to map learned features to the distributions of unseen modalities.

Table 8: Comparison of predictive power vs. transfer capability.

| Training → Test Bands | x-cashew-plantation | | x-SA-crop-type | |
| | DOFA | DINOv3 | DOFA | DINOv3 |
| --- | --- | --- | --- | --- |
| *In-Distribution (Upper Bound)* | | | | |
| S2 → S2 | 80.26 | 78.89 | 33.40 | 31.30 |
| RGB → RGB | 79.97 | 78.55 | 26.69 | 32.32 |
| N'S$_1$S$_2$ → N'S$_1$S$_2$ | 79.97 | 79.54 | 26.49 | 27.38 |
| S1 → S1 | 65.60 | 67.91 | 17.96 | 19.55 |
| *Cross-Band Transfer* | | | | |
| RGB → S1 | 6.27 | 6.78 | 5.21 | 5.07 |
| S2 → S1 | 6.14 | 6.31 | 5.19 | 6.30 |

# H  WAVELENGTH SENSITIVITY

To investigate if models rely on specific spectral signatures or wavelength-independent spatial structures, we fine-tuned a "color-blind" DINOv3 where embedding weights are shared across all

input bands. As shown in Table 9, while the color-blind model sees a drop in *In-Distribution* performance (RGB→RGB), it significantly improves generalization to the distinct SAR modality (RGB→S1). This suggests that enforcing wavelength insensitivity encourages the model to rely on geometric and textural features that are more robust to domain shifts.

Table 9: Comparison of standard vs. "color-blind" DINOv3 (shared embedding weights). The color-blind model improves zero-shot transfer to SAR (S1) at the cost of in-distribution accuracy.

| Dataset | Model | In-Dist. RGB→RGB | No-Overlap RGB→S1 | No-Overlap RGB→N'S$_1$S$_2$ |
|---|---|---|---|---|
| x-so2sat | DINOv3 | $60.83 \pm 1.66$ | $5.66 \pm 0.55$ | $11.03 \pm 1.84$ |
| | color-blind DINOv3 | $54.32 \pm 2.74$ | $8.32 \pm 1.41$ | $17.26 \pm 2.51$ |
| x-SA-crop-type | DINOv3 | $32.32 \pm 0.38$ | $5.07 \pm 0.01$ | $18.03 \pm 0.22$ |
| | color-blind DINOv3 | $26.17 \pm 0.44$ | $5.09 \pm 0.02$ | $19.98 \pm 0.87$ |
| x-harvey-flood | DINOv3 | $70.75 \pm 3.70$ | $28.79 \pm 4.82$ | $40.73 \pm 8.93$ |
| | color-blind DINOv3 | $69.09 \pm 3.86$ | $43.48 \pm 3.21$ | $29.59 \pm 3.52$ |

# I  COMPUTE RESOURCES

Our experiments were performed on three machines: DGX A100 and DGX H100 and one HGX H100 node kindly donated by Nebius.ai cloud.

The final pretraining of $\chi$ViT required 12 days of 8 H100s (96 H100-days). Before the final version we had one more similar run which had a bug in the layer unfreezing code which resulted in a poor performance.

Fine-tuning compute strongly depends both on the model and the dataset. We used 5 seeds for every pair. We also performed hyperparameter search on 4 datasets and 2 models with 27 or 48 combinations of hyperparameters. All these experiments were scheduled with Slurm on A100 and H100 nodes, and we did not track which experiments went to which GPU. In total, we estimate all fine-tuning efforts (including hyperparameter search) used 45 GPU-days.

We also estimate that we wasted another 40 GPU-days on running initial experiments for each baseline model. Many experiments were performed on older versions of the datasets (e.g. original BigEarthNet v1.0, or non-GeoBench versions of datasets) that were excluded from this paper.

