# OpenReview forum: "GeoCrossBench: Cross-Band Generalization for Remote Sensing"
_ICLR.cc/2026/Conference — Submitted to ICLR 2026_

### Official Review · Reviewer_gfdj · 2025-10-30

**Soundness:** 2
**Presentation:** 2
**Contribution:** 2
**Rating:** 6
**Confidence:** 3

**Summary:**

The paper introduces GeoCrossBench, an extension to the GeoBench benchmark designed for evaluating the cross-satellite generalization of remote sensing models. A new evaluation protocol is introduced that measures model performance in-distribution, on satellites with no overlapping bands, and on satellites with additional bands compared to training data. The work also presents χViT, a self-supervised extension of ChannelViT, aimed at improving cross-satellite transferability.

**Strengths:**

- Generalization across diverge satellite image sensors is a very important problem that could unlock the use of diverse data and improve the use of newly launched satellites from the get go, this paper advances research towards this important direction by releasing a dataset and proposing an evaluation framework.
- The evaluation framework presented is sensible and covers train-test discrepancy by separating cases to in-distribution, no-overlap bands and superset of bands which makes sense from an application perspective.

**Weaknesses:**

- The limitation of >100M parameter models seems quite restrictive for the task at hand since it excludes the best performing foundation models.
- The proposed evaluation tasks are limited but can be expanded in followup works.
- The value of this work depends greatly on the quality of the proposed dataset, it would be nice to see some samples as part of this submission but none were included.
- Figure captions are not sufficient for understanding the figures, they should be expanded to include all relevant information without requiring the reader to go through the main text.
- Related work should include works on out-of-distribution performance of deep neural networks and machine learning for remote sensing.

(Minor minor)
- typo in l.114 "challange"

**Questions:**

- Can you explain what are the performance metrics presented in Table 2 and what the * symbol represents?

---

> ### Author Response · Authors · 2025-11-21
> **The limitation of >100M parameter models**
>
> We acknowledge that restricting our evaluation to <100M parameters excludes larger foundation models. However, this decision was necessary to ensure a rigorous evaluation given our computational constraints. Running comprehensive experiments with multiple random seeds and diverse downstream tasks on >100M parameter models was not feasible within our current compute budget.
>
> Could you clarify which specific >100M models would be the most relevant? While we cannot run new extensive benchmarks during the rebuttal period, we will ensure they are properly cited and discussed as a necessary direction for future scaling studies.
>
> (Please note that the other reviewer suggested to consider TerraMind. We are now running the experiments and will include the results in the next revision during this discussion period.)

---

> ### Author Response · Authors · 2025-11-21
> **Figure Captions Are Not Sufficient**
>
> We appreciate this feedback. We will revise the figure captions to ensure they are more self-explanatory. We will pay special attention to Figure 3, expanding its caption to clearly summarize the results and present the key takeaways directly.

---

> ### Author Response · Authors · 2025-11-21
> **Related work should include works on out-of-distribution performance of deep neural networks and machine learning for remote sensing**
>
> Thanks for suggesting further related work on out-of-distribution performance more generally beyond our focus of cross-satellite and cross-band generalization. We are working on this, and have so far identified https://arxiv.org/abs/2507.13385 as relevant, and we will include more references on this topic before the end of the discussion period.

---

> ### Author Response · Authors · 2025-11-21
> **Q: Can you explain what are the performance metrics presented in Table 2 and what the * symbol represents?**
>
> Thanks for pointing out this.
>
> 1. The * symbol indicates that the backbone was frozen during fine-tuning. We will revise the text and caption to indicate this.
> 2. The performance metrics of each setting reported in Table 2 represent the average scores across all GeoCrossBench datasets. For each individual dataset, we used the corresponding standard metric as detailed in Section 2.3. Since all selected metrics share a common bounded range, averaging them provides a statistically reasonable summary of the model's generalizability across the benchmark.
>
> We will update the caption of Table 2 to ensure these details are clearly stated.

---

### Official Review · Reviewer_hRxA · 2025-11-01

**Soundness:** 3
**Presentation:** 4
**Contribution:** 3
**Rating:** 6
**Confidence:** 2

**Summary:**

This paper introduces GeoCrossBench, an extension of the GeoBench benchmark for evaluating generalization capabilities of remote sensing foundation models across satellites with differing spectral characteristics. The benchmark defines three evaluation protocols: (a) In-distribution performance, (ii) Generalization to satellites with no band overlap, and (iii) Generalization to satellites with additional bands compared to the training set. The authors also propose ChiViT, a self-supervised extension of ChannelViT, designed to improve cross-satellite transfer by encouraging robustness to variations in input spectral bands. Through extensive experiments, the authors find that even strong foundation models (DOFA, TerraFM) fail to outperform general-purpose models (e.g., DINOv3) in-distribution; that performance drops substantially (2-4x) when generalizing to unseen satellites; and that ChiViT yields the best results in the no-overlap setting. They further show that fine-tuning only the last linear layer using oracle labels can partially recover performance, indicating that cross-satellite generalization remains an open challenge.

**Strengths:**

1. The paper is well written and interesting
2. The considered problem (cross-satellite generalization for remote sensing) is relevant as Earth observation systems diversify rapidly.
3. The proposed self-supervised, band-sampling extension to ChannelViT is an interesting approach to improve transferability.

**Weaknesses:**

1. Improvements over DinoV3 are not significant
2. Metrics between methods show an important variance, making the results difficult to read

**Questions:**

**Major comments**
1. An interesting part of the paper is this paragraph : "One of the reasons for the relatively strong performance of ChiViT compared to other multispectral models might be the trick of sampling of the bands during pretraining. The models might learn to rely less on band-specific features and instead focus on patterns shared across bands, which then improves cross-band generalization performance. Sampling of channels during fine-tuning might also be beneficial." Sadly, this is not tested. Could the authors add a vizualization, or experiments, that could strengthen these claims? I think this would add greatly to the paper.

**Minor comments**
1. The goal of the paper is to demonstrate that the proposed architecture / training strategy shows better transfer ability than existing approaches. In this context, I believe that Table 2 is unclear. All metrics are very similar, the coloring seems to be global (e.g. column 1 is compared to column 5, is it meaningful?) where I would assume that the performance is only evaluated for one setup (i.e. within a column, and not between columns). There, the most important columns are those in "No-overlap", and indeed, the authors report a better metric than concurrent methods. At the moment, this column is barely stressed, due to the color scheme pointing more towards the in-distribution + superset approaches...

---

> ### Author Response · Authors · 2025-11-21
> **Major Comment 1**
>
> Thanks for highlighting this. For the rebuttal we experiment with sampling during fine-tuning as suggested. We fine-tune ChiViT with channel subsampling on three datasets. The results are inconsistent: on x-bigearthnet subsampling helps in the No-Overlap setting, but on x-SA-crop-type and x-harvey-flood it is only better in one setting. We will add this table to Appendix and will highlight that unlike the results in ChannelViT paper, the Hierarchical Channel Sampling can help or hurt during fine-tuning in our benchmark settings.
>
> | ChiViT | RGB → RGB | S2 → S2 | RGB → S1 | S2 → S1 | RGB → N'S1S2 | RGB → RGBN | S2 → S2+S1 |
> | :--- | :--- | :--- | :--- | :--- | :--- | :--- | :--- |
> | **x-SA-crop-type**`(sampling)` | 24.04±0.91 | 30.99±1.07 | 5.13±0.07 | 5.07±0.00 | 23.10±0.57 | 24.46±0.97 | 29.49±0.97 |
> | **x-SA-crop-type**`(regular)` | 19.58±1.28 | 30.85±0.54 | 5.12±0.09 | 5.09±0.02 | 19.67±2.16 | 20.08±1.17 | 29.49±0.42 |
> | **x-harvey-flood**`(sampling)` | 70.66±0.94 | 45.72±10.68 | 10.59±6.40 | 3.64±1.77 | 46.90±2.47 | 59.70±2.29 | 40.98±8.08 |
> | **x-harvey-flood**`(regular)` | 64.30±7.87 | 56.22±12.07 | 4.11±3.18 | 10.43±4.80 | 18.01±6.86 | 59.24±6.08 | 51.94±12.43 |
> | **x-bigearthnet**`(sampling)` | 67.39±0.59 | 70.85±0.49 | 6.83±2.49 | 16.95±4.72 | 21.88±13.49 | 66.12±0.79 | 69.49±1.91 |
> | **x-bigearthnet**`(regular)` | 71.03±0.88 | 73.15±0.43 | 5.14±1.29 | 6.05±1.37 | 19.34±3.42 | 68.31±1.18 | 70.91±0.55 |

---

> ### Author Response · Authors · 2025-11-21
> **Minor Comment 1**
>
> We appreciate this feedback. We will improve our coloring of Table 2 by applying color maps within each setting to better highlight variations in models’ performance.
>
> We would like to also clarify that while proposing a training strategy is indeed a major goal of this paper, the other major goal is encouraging future work on cross-satellite and cross-band transfer of remote sensing models by designing the proposed benchmark and evaluation setup. In this way our experiments are also empirically informative about existing models and how well they transfer.

---

### Official Review · Reviewer_3Eev · 2025-11-04

**Soundness:** 2
**Presentation:** 3
**Contribution:** 2
**Rating:** 2
**Confidence:** 5

**Summary:**

GeoCrossBench introduces a benchmarking paradigm (Fig. 1) that evaluates geospatial foundation models (GFMs) on their capacity to fine-tune on classification, semantic segmentation, and change detection downstream tasks such that at those GFMs are tolerant to an input of different spectral bands. The work builds on data from GEO-Bench (Lacoste et al., 2023) by picking those with Sentinel-2 imagery and pairing these with Sentinel-1 (GRD product), cf. Tab. 1. Moreover, the authors provide a benchmark baseline model (Tab. 2) by pre-training a ChannelViT on optical and radar remote sensing imagery at multiple spatial resolutions, cf. Tab. 3. This baseline, termed ChiViT, along with general purpose vision models DINOv2/3 outperform the GFMs considered for testing, Tab. 2.

**Strengths:**

The paper is written in plain English, with figures and tables underlining the findings from the experiments conducted. The appendix provides additional details supporting reproduction of the work.

**Weaknesses:**

# Soundness
- 2: fair

# Presentation
- 3: good

# Contribution
- 2: fair

# Strengths
The paper is written in plain English, with figures and tables underlining the findings from the experiments conducted. The appendix provides additional details supporting reproduction of the work.

# Weaknesses
I appreciate the author's effort towards benchmarking the utility of GFMs. However, the current manuscript resembles just a slight variation of work that has been previously published under the same GeoCrossBench name.

As domain expert in remote sensing I rate the setup of cross-band validation through downstream tasks somewhat problematic given that the bandwidth of a spectral channel is physically relevant. For example biomass (cf. `x-cashew-pantation` and `x-SA-crop-type`) is sensitive to the (normalized) difference of the red and the near-infrared channels. Randomly replacing RGB-channel-fine-tuned inputs with random channels in the invisible range, including wavelength as far off as C-band radar is a bold step that requires careful ablation studies to understand the model's behavior, cf. some questions related below. However, I agree that domain-specific GFMs should meet the expectations to outperform general vision models not tuned towards Earth observation such as DINOv3.

Given the two arguments above, I hesitate to recommend this work for publication at ICLR at its current state: a significant part of the presented work has been published elsewhere, and the benchmark design protocol raises questions about the physical interpretation. From that angle I suggest you avoid terms such as (l19-20):
> First, we show that even the *best foundation* models for remote sensing (DOFA, TerraFM) [...]

**Questions:**

- Did you considered benchmarking the successor of Prithvi, TerraMind?
- Would you please elaborate on huge jump in performance for DINOv3 frozen vs. full fine-tuning in Tab. 2?
- How do you explain the DOFA *anomaly* where the performance drops on fine-tuning for the _No-Overlap_ scenario?
- To improve the quality of the current manuscript, pls consider ablation studies why ViTs work well when trained insensitive to the wavelength of a satellite sensor? Why does texture and spatial structure seem sufficient?
- By when will you share GeoCrossBench's data, code, and models under which license?

**Details Of Ethics Concerns:**

Previously published work with significant overlap (some sentences close to copy-paste) of accepted (https://openreview.net/group?id=ICML.cc/2025/Workshop/TerraBytes#tab-accept-without-proceedings) manuscript at 2025 ICML *TerraBytes* workshop: https://openreview.net/pdf?id=7MNndxX1Xq , the paper states:
  > On publication we will share the GeoCrossBench data, code, and models.

  which I was unable to identify on web search, yet. To me, it raises the question when and how the authors will make available these assets.

Updates compared to the 2025 ICML workshop article :
    * add novel geo-foundation model TerraFM, and latest DINOv3 vision model
    * slightly extended evaluation protocol for transfer to situation with more bands than during pre-training
    * novel visualization of results

---

> ### Author Response · Authors · 2025-11-21
> **The current manuscript resembles just a slight variation of work that has been previously published under the same GeoCrossBench name**
>
> We respectfully ask that the reviewer evaluate this work on its own, per the ICLR 2026 policies, and not in relation to an unpublished workshop paper. Here is the relevant quote from the Dual Submission Policy section: “papers that have appeared on non-peer reviewed websites (like arXiv) or that have been presented at workshops (i.e., venues that do not have publication proceedings) do not violate the policy.”
>
> We thank the reviewer for paying attention to this matter, and welcome their expert feedback on this work.

---

> ### Author Response · Authors · 2025-11-21
> **Concern regarding Physical Relevance of Spectral Channels in Cross-Band Validation (x-cashew-plantation and x-SA-crop-type)**
>
> Thanks for highlighting these two datasets. Indeed, the tasks related to biomass prediction are known to be correlated with NIR and RED channels from the remote sensing literature. On the other hand, less is known on how much relevant information is contained in other bands, and a purpose of the proposed benchmark is to evaluate potential transfer and generalization across bands. To more specifically empirically measure this, we performed experiments on these two datasets in the in-distribution setting with different band combinations using DOFA and DINOv3. The results are in the table below:
>
> - both RGB and Near-to-short-wave-infrared wavelengths have as much predictive power as the full spectrum of Sentinel-2 in x-cashew-plantation, and ~20% less in x-SA-crop-type.
> - C-band SAR has 20% less information in x-cashew-plantation and ~40% less in x-SA-crop-type compared to the full spectrum of Sentinel-2 (as measured by the respective metrics of the datasets).
> - Still, the performance of the transfer from RGB/Sentinel-2 to C-band SAR by all models is significantly worse than the predictive power of C-band SAR (e.g. 6.27 vs. 65.60 on x-cashew-plantation), so there is a lot of room for improvements for foundation models.
>
> | | `x-cashew-plantation` | `x-cashew-plantation` | `x-SA-crop-type` | `x-SA-crop-type` |
> | :--- | :---: | :---: | :---: | :---: |
> | **Training → Test Bands** | **DOFA** | **DINOv3** | **DOFA** | **DINOv3** |
> | S2 → S2 | 80.26 | 78.89 | 33.40 | 31.30 |
> | RGB → RGB | 79.97 | 78.55 | 26.69 | 32.32 |
> | N’S1S2 → N’S1S2 | 79.97 | 79.54 | 26.49 | 27.38 |
> | S1 → S1 | 65.60 | 67.91 | 17.96 | 19.55 |
> | RGB → S1 | 6.27 | 6.78 | 5.21 | 5.07 |
> | S2 → S1 | 6.14 | 6.31 | 5.19 | 6.30 |
>
> We are going to include this table in the Appendix in the next revision.
>
> One experiment below sheds more light on how much is wavelength specificity important. Nevertheless, good foundation models should be able to perform closer to the in-distribution performance on SAR.

---

> ### Author Response · Authors · 2025-11-21
> **"First, we show that even the best foundation models for remote sensing (DOFA, TerraFM)"**
>
> Thanks a lot for pointing out this issue. We wanted to highlight the top two remote-sensing-specific models according to our experiments. We decided to simply remove the names in the parentheses to avoid confusion.
>
> Please let us know if we misunderstood the suggestion.

---

> ### Author Response · Authors · 2025-11-21
> **Q: Did you considered benchmarking the successor of Prithvi, TerraMind?**
>
> Thank you for the interest in the very latest models and successors of Prithvi in particular. Prithvi 2.0 doesn’t have a ViT-B size, so we didn’t include it in our comparison table.
>
> For TerraMind, we respectfully note that it is concurrent work and was only published at ICCV 2025 after the ICLR 2026 deadline. Nevertheless, in the spirit of an informative response we have initiated experiments for TerraMind using 3 random seeds, and preliminary results are now available for several datasets. Initial results show the model has robust performance when fully fine-tuned, with strong results in classification but lower performance in segmentation and change detection. By the time the discussion period ends we will have the full results and will include TerraMind in the main tables and figures of the next revision.
>
> | Dataset | RGB | RGB (frozen) | N’S1S2 | N’S1S2 (frozen) | VV VH | VV VH (frozen) |
> | :--- | :---: | :---: | :---: | :---: | :---: | :---: |
> | **x-bigearthnet** | 94.46±0.11 | 93.07±0.03 | 89.61±0.13 | 88.88±0.10 | 90.18±0.17 | 83.18±0.12 |
> | **x-eurosat** | 96.60±0.37 | 85.63±0.09 | 21.30±2.05 | 27.60±0.08 | 20.70±1.79 | 15.93±0.33 |
> | **x-harvey-flood** | 48.67±1.52 | 34.76±0.62 | 32.14±1.31 | 12.72±0.41 | 21.95±1.30 | 1.00±0.53 |
> | **x-harvey-building** | 35.74±0.26 | 34.11±1.22 | 12.08±1.56 | 16.99±2.00 | 24.73±2.01 | 17.35±2.59 |

---

> ### Author Response · Authors · 2025-11-21
> **Q: Would you please elaborate on huge jump in performance for DINOv3 frozen vs. full fine-tuning in Tab. 2?**
>
> The difference between frozen and full fine-tuning results can be large. When reporting this we average across datasets and settings and find a similar gap for other models. DINOv3 is not the most extreme, and Satlas and Prithvi have bigger jumps (10 vs. 7 points!). Please check Figure 4a in Appendix B for more.
>
> When diving into details, we see that DINOv3-frozen struggles less with RGB bands and struggles more with the non-RGB bands (compared to full fine-tuning). This is expected, as the pretrained DINOv3 has never seen non-RGB bands.

---

> ### Author Response · Authors · 2025-11-21
> **Q: Pls consider ablation studies why ViTs work well when trained insensitive to the wavelength of a satellite sensor? Why does texture and spatial structure seem sufficient?**
>
> Thanks for your suggestion. To test this the impact of wavelength sensitivity in our benchmark we additionally fine-tune DINOv3 with a shared embedding layer across all bands, so it insensitive to wavelength and processes all bands in the same way.  For this rebuttal experiment we use x-so2sat(classification), x-SA-crop-type(segmentation) and x-harvey-flood(change detection) datasets.
>
> - In the *In-Distribution* setting the performance is worse. This result suggests that sharing the embedding layer parameters across bands reduces the model's capacity to specialize to the training modalities.
> - In the *No-Overlap* setting the performance improves, except in one case. We hypothesize that having the same embedding for all bands regularizes the model, preventing specialization to specific bands, and encourages generalization to testing bands by forcing the sharing of features across all of the training bands.
>
> | Color-blind DINOv3 | RGB→RGB  | RGB → S1 | RGB → N’S1S2 |
> | :---- | :---- | :---- | :---- |
> | x-so2sat | 54.32±2.74 | 8.32±1.41 | 17.26±2.51 |
> | x-SA-crop-type | 26.17±0.44 | 5.09±0.02 | 19.98±0.87 |
> | x-harvey-flood | 69.09±3.86 | 43.48±3.21 | 29.59±3.52 |
>
> | DINOv3 | RGB→RGB | RGB → S1 | RGB → N’S1S2 |
> | :--- | :--- | :--- | :--- |
> | x-so2sat | 60.83 ± 1.66 | 5.66 ± 0.55 | 11.03 ± 1.84 |
> | x-SA-crop-type | 32.32 ± 0.38 | 5.07 ± 0.01 | 18.03 ± 0.22 |
> | x-harvey-flood | 70.75 ± 3.70 | 28.79 ± 4.82 | 40.73 ± 8.93 |

---

> ### Author Response · Authors · 2025-11-21
> **Q: By when will you share GeoCrossBench's data, code, and models under which license?**
>
> As is common practice at machine learning venues like ICLR we will share the data, code, and models on publication = at the time of the conference with standard licenses (BSD-2 for code, CC-BY-4.0 for data, etc.).

---

### Author Response · Authors · 2025-12-02
**Summary of Updates in the Revised Version**

We have uploaded a revised version of our paper including the reviewers' feedback. The key updates are as follows:
- [3Eev] To address concerns regarding the x-cashew-plantation and x-SA-crop-type datasets, we added Appendix G (Table 8) to empirically estimate the predictive power of different band combinations.
- [3Eev] In response to the suggestion to benchmark TerraMind and Prithvi 2.0 models, we added TerraMind to our set of baseline models and performed all the necessary experiments. The updated results are now included in Table 2, as well as Figures 3 and 4. We did not include Prithvi 2.0 as it does not offer a ViT-B variant, which is required for consistency with our other <100M parameter baselines.
- [3Eev] Following the reviewer's suggestion, we included a new ablation study in Appendix H to test the impact of wavelength sensitivity on the benchmark.
- [3Eev] We removed the top two remote-sensing-specific model names from the abstract parentheses to avoid confusion.
- [hRxA] We added Appendix F to analyze the impact of channel sampling during fine-tuning stage.
- [hRxA] We improved Table 2 by applying color maps within each evaluation setting to better highlight variations in model performance.
- [gfdj] We updated figure and table captions throughout the paper to ensure they are self-explanatory.

**Note on the Ethics Flag**: We would also like to direct the new AC's attention to the [thread regarding the ethics flag](https://openreview.net/forum?id=ZpenqZE0yL&noteId=8dSZLc4dKj) raised by Reviewer 3Eev. This concerns the confusion of the reviewer regarding the ICLR policy on non-archival workshop papers. The issue was discussed with the previous AC.

---

### Meta-Review · Area_Chair_VVfR · 2025-12-28

**Summary:**

Reviewer 3Eev raises a concern about potential dual submission. Based on the information available, I conclude that the paper does not violate the dual-submission policy: the earlier version was presented at a workshop in a non-archival form (i.e., not published in the proceedings) and appears to have been a presentation-only submission. Under these conditions, submitting the work to ICLR is allowed. I therefore did not consider this comment in my final decision.

Because the reviewers expressed relatively low confidence in their assessments (hRxA: 2; gfdj: 3), I conducted my own review and the final recommendation is based on summarizing my own assessment and the other reviewers’ concerns.

On the one hand, the paper tackles an important question: how well do existing geospatial foundation models (GFMs) generalize across satellite image sensors that differ substantially. On the other hand, I find that GeoCrossBench has notable weaknesses in dataset selection, evaluation design, and its description of the new ChiViT model.

Regarding dataset selection, most of GeoCrossBench (6 of 10 datasets) is drawn from GeoBench, while the four newly introduced datasets are substantially smaller (fewer than 500 training samples each, versus more than 1,000 training samples for the GeoBench datasets). In addition, the authors augment GeoBench with Sentinel-1 SAR data but do not provide a quality assessment of the fusion process—for example, whether tiles from different sensors are accurately aligned. More broadly, the paper does not include a data-contamination analysis, which is particularly important for foundation-model benchmarks: the GFMs’ pretraining data may already contain some of the GeoCrossBench imagery, or even related labels. Echoing Reviewer gfdj, the value of this work depends heavily on the quality of the proposed dataset, and I remain concerned about the dataset’s reliability.

Regarding the evaluation design, Table 2 reports point estimates without error bars. Given the small dataset sizes, uncertainty estimates are important for interpreting unexpected findings—for example, DOFA’s performance drop under full fine-tuning, as noted by Reviewer 3Eev. In addition, the paper applies inconsistent hyperparameter optimization (HPO) across datasets. Appendix E.1 states that grid search is used for x-sen1floods11, x-harvey-building, x-harvey-flood, and x-oscd, while a fixed learning rate is used for the remaining datasets. Because the numbers in Table 2 are averaged across datasets, a consistent HPO protocol is necessary to avoid introducing unnecessary variance—closely related to Reviewer hRxA’s concern about metric variability. Finally, as a benchmark, the paper does not clearly specify how results should be reported on GeoCrossBench (e.g., recommended splits, repeats/seeds, and a good summary metric).

For ChiViT, the training recipe remains unclear to me even after reading Section 3.2 and Appendices C and D. While Table 4 presents ablations on design choices such as “Shared Subset Sampling,” “Shared Projection,” “Shared Head,” and “Parallel Data Coefficient,” these terms are not sufficiently defined, making it difficult to understand what is being changed and why. Clearer descriptions of each component and the overall training pipeline would materially improve reproducibility. Finally, I am not convinced the comparison is fully fair: ChiViT seems to be designed for cross-band transfer, so its evaluation alongside more general-purpose GFMs may conflate benchmark performance with task-specific architectures.

Based on these concerns, I think the paper does not meet the bar of ICLR and recommend for rejection. The paper could be strengthened by narrowing its scope—either focusing on GeoCrossBench as the primary contribution, or centering the work on ChiViT—so that the methodology, experiments, and claims can be developed in greater depth and with clearer takeaways.

**Reviewer Concerns:**

I think the reviewer hRxA and gfdj have low confidence and did not express the concerns clearly. On the other hand, Reviewer 3Eev's concern on dual submission is not valid. See Summary for other concerns for the paper.

**Reviewer Scores:**

I think the scores will change dramatically if we have the full discussion. The reviewers either have low confidence or is voting based on false concerns on dual submission.

---

### Decision · Program_Chairs · 2026-01-26

Reject